# Comparison of a Wearable Accelerometer/Gyroscopic, Portable Gait Analysis System (LEGSYS+^TM^) to the Laboratory Standard of Static Motion Capture Camera Analysis

**DOI:** 10.3390/s23010537

**Published:** 2023-01-03

**Authors:** Ryan Homes, Devon Clark, Sina Moridzadeh, Danijel Tosovic, Wolbert Van den Hoorn, Kylie Tucker, Mark Midwinter

**Affiliations:** 1School of Biomedical Sciences, The University of Queensland, Brisbane, QLD 4067, Australia; 2School of Health and Rehabilitation Sciences, The University of Queensland, Brisbane, QLD 4072, Australia; 3ARC ITTC Joint Biomechanics, Queensland Unit for Advanced Shoulder Research, Movement Neuroscience Group, Injury Prevention Group, Exercise & Movement Science, School of Exercise and Nutrition Sciences, Queensland University of Technology, Brisbane, QLD 4067, Australia

**Keywords:** walking, gait, motion capture, LEGSYS+^TM^, phase parameters

## Abstract

Examination of gait patterns has been used to determine severity, intervention triage and prognostic measures for many health conditions. Methods that generate detailed gait data for clinical use are typically logistically constrained to a formal gait laboratory setting. This has led to an interest in portable analysis systems for near clinical or community-based assessments. The following study assessed with the wearable accelerometer/gyroscopic, gait analysis system (LEGSYS+^TM^) and the standard of static motion capture camera (MOCAP) analysis during a treadmill walk at three different walking speeds in healthy participants (n = 15). To compare each speed, 20 strides were selected from the MOCAP data and compared with the LEGSYS+ strides at the same time point. Both scatter and bland-Altman plots with accompanying linear regression analysis for each of the parameters. Each stride parameter showed minimal or a consistent difference between the LEGSYS+ and MOCAP, with the phase parameters showing inconsistencies between the systems. Overall, LEGSYS+ stride parameters can be used in the clinical setting, with the utility of phase parameters needing to be taken with caution.

## 1. Introduction

In the clinical setting, gait patterns have been extensively studied in relation to a diverse range of health conditions. Assessment of walking speed is the most commonly used gait test and has good reliability [1,2] and clinical validity [3,4] across multiple diseases. Walking speed tests are often used as prognostic measures for health conditions such as chronic obstructive pulmonary disease [5,6,7] or pulmonary hypertension [8,9,10]. Furthermore, decreased walking speed was highlighted by Fried (2001) as an important functional feature of the frailty phenotype [11,12,13]. For example, elderly or frail individuals who walk slow (<1 m/s) are at a higher relative risk to be in institutionalized care or hospitalized [14,15]. Speed tests are simple and easy to perform in the clinical or community setting without the requirement of a laboratory based detailed gait assessment system. However, other potentially important elements of gait are ignored. More detailed gait.

Parameters may give mechanistic insights and earlier indications of functional deficits that may help target early interventions [16,17,18,19].

A recent systematic review and metanalysis concluded that certain gait parameters are the critical gait metrics most associated with a propensity to developing a physical disability, suffering from falls, becoming dependent and institutionalised, and mortality. These are commonly referred to as ‘major adverse events’ within the literature [20,21]. The gait cycle can consist of parameters related to a stride and parameters related to phases within a stride. Stride measures include velocity [22,23,24], stride length [22,23,24,25] and cadence (steps/minute) [24,25,26] (Figure 1). Phase measures include leg swing, leg stance and single or double limb support [27]. Taken together, stride and phase gait parameters could aid in early identification and monitoring of the degree of frailty. This is important as the prevalence of frailty is increasing, and is expected to reach 26–28% of the total Australian population by 2051 [28]. 

The gait cycle (Figure 1) is divided into stance (approximately 60% of cycle time) and swing (approximately 40% of cycle time) phases for each limb [29]. Each swing and stance phase coincides with 8 functional periods along with periods of double-limb support (DLS) [30,31]. Right leg is black, and left leg is white. From left to right, the first stage in the cycle starts with the initial contact of the foot (in this case the right), striking the ground and moving into a loading response (yellow boxes). This period constitutes the start of the stance phase (60%) and the first period of double limb support (DLS: 10–12%). This is followed by a transition into the mid and terminal stance phase, leading into the right foot starting the swing phase (dark blue boxes). This pre-swing period aligns with the second instance of DLS. After the conclusion of the DLS the initial, mid and terminal swing phases of the right foot begin, which align with the right swing phase (40%). The image is adapted from [28,29]. Stride length is determined from one heel strike to a consecutive heel strike on the same side, stride time is the duration in between these heel strikes velocity is determined as stride length/time from one heel strike to a consecutive heel strike on the same side.

A variety of laboratory-based methods can be used to collect reliable and reproduceable gait data but have limited usability outside of the laboratory. Measurement of gait outside a laboratory or clinical environment is important as changes in gait pattern have been shown when an examination is performed within a gait laboratory compared to a familiar environment [32]. The wearable solution addresses this important issue, allowing collection of data in a real-world setting, in an environment in which the subject will be functioning. Ninety-eight percent of elderly people in Australia live in a community setting. In order to allow frail individuals to remain living in their homes rather than entering residential aged care (the ‘ageing in place’ scheme) in Australia, there needs to be consideration of clinical evaluations performed local in individuals’ homes [28,33,34]. As a result, gait analysis systems that are both cost-effective and useable outside the laboratory, have been developed [35]. This includes wearable systems, that are small in size and lightweight with sensors located at the ankles, thighs and waist, to track movements [35,36]. These systems have been employed in assessing gait abnormalities in osteoarthritis [37,38,39], vestibular function degradation [40] and Parkinson’s disease [41,42] but their comparison to static motion capture camera analysis (MOCAP) is not well documented.

The objective of this study was to assess the validity of several stride and phase gait parameters derived from a portable gait analysis system (LEGSYS+^TM^, Locomotion Evaluation and Gait System) against MOCAP at different walking speeds, before utilising the LEGSYS+^TM^ (Biosensors, Cambridge, MA, USA) in future clinical use (e.g., outpatient clinics and GP surgeries) and community-based studies.

## 2. Materials and Methods

### 2.1. Participants

Fifteen participants aged between 18–59 years (median = 24.5, IQR = 21–29, height: 173.7 ± 10.2, weight: 79.5 ± 19.8, 6 male, 9 female) volunteered for this study. Participants were excluded if they: were <18 years of age, experienced mobility limiting injuries or diseases in the last use of mobility aids (e.g., walking stick or walking frame), unintentional weight loss of 4.5 kg in the last 12 months, and/or reported a history of health conditions that may affect walking (such as chronic fatigue, osteoporosis, muscular dystrophy, neuromuscular disease, malignancy, previous surgery or extensive damage to the leg or foot). Assessments were conducted in accordance with a protocol and consent processes approved by The University of Queensland Human Ethics Committee (201900959).

### 2.2. Experimental Setup

Phase gait parameters were simultaneously collected by the LEGSYS+^TM^ and MOCAP at varying walking speeds while walking on a treadmill (Trimline T345). Four reflective non-collinear marker clusters were placed on the heels of the left and right shoes of standard testing plimsoll (volleys) type (Figure 2). A 12-camera (motion capture) 3D optical movement registration system (OptiTrack Flex 13) was used to record movements of participant’s feet, using Motive software (Natural Point, OR, USA), sampled at 120 samples/s. The LEGSYS+^TM^, consisting of five Bluetooth-enabled inertial measurement units (consisting of 3D gyroscopes and 3D accelerometers, sampled at 100 samples/s), was paired to the LEGsys software program before placing the sensors on the participant. The LEGSYS+^TM^ was then fitted to the participant in standardised positions as follows (Figure 3); first, leg length (distance between the lateral femur epicondyle to the floor), and thigh length (distance between lateral femur epicondyle and the anterior superior iliac spine) were measured, whilst the participant was standing upright in plimsoll shoes. Then, the leg motion sensors were secured with adjustable straps, placed at 30% of the leg length, distally from the lateral femoral epicondyle (i.e., on the lower leg). The thigh sensor was placed 30% of the thigh length, proximally from the lateral femoral epicondyle (i.e., on the upper leg), and the waist sensor was placed in the midline posteriorly, at the level of the umbilicus. Participants faced positive Z-axis (anterior-posterior, forward and backward) positive X-axis was to the left (medial-lateral, side to side movement), and positive Y-axis was upwards (superior-inferior, up and down). A continuous recording was then performed on the treadmill at walking speeds of 2, 3 and 4 km/h, while simultaneous collected by both systems. Walking speed was increased at one-minute intervals [43,44,45].

### 2.3. Stride Comparison between Systems

The following gait parameters were extracted from the MOCAP data for comparison against LEGSYS as described in the LEGSys User Manual (stride time, stride length, stride velocity, cadence, left/right swing, left/right stance and left/right double support phase). All gait parameters were based on heel strike and toe off times. Using the MOCAP data, strides were determined using definitions for Heel strikes (HS) and toe-off (TO) using previously stated methods [46]. Briefly, left and right heel strikes were determined from the local minima of the respective mean heel cluster vertical axis position. Toe off was determined as the peak of the vertical velocity determined as the time differentiated mean vertical position of the respective cluster after heel strike. Then, left and right stride time was determined as the time in between consecutive heel strikes on the same side. Stride length was determined by multiplying stride time by the ‘actual’ treadmill speed. The remaining parameters (cadence, stride velocity, stance, swing and double support) were generated using the calculated stride time and length. At each treadmill speed, the last 20 complete stride cycles before the speed change were used (Figure 1).

There might be a discrepancy between set and actual treadmill speed. To measure the actual treadmill speed, the average forward-backward position of the right heel cluster was filtered using a low-pass second order bi-directional Butterworth filter with a cut-off frequency at 5 Hz (bi-directional filter design reported) [47,48]. Then, mean position of the cluster was differentiated over time to determine velocity. A short time period of the forward backward cluster velocity from 0.25–0.42 s after each of the 20 included heel strikes was extracted. This time period coincided with the small part of the stance phase approximately in between the heel strike and consecutive toe-off. In this period the foot travels at the speed of the treadmill belt. Actual treadmill speed was then determined as the mean across all save time points.

To ensure that the same 20 strides were compared between the LEGSYS+ and MOCAP system, cross correlation between the calculated stride time from MOCCAP and LEGSYS+ was performed (see a participant example in Figure 4).

### 2.4. Statistical Analysis

Matlab (v. R2020b, Mathworks, Natick, MA, USA) was used for the statistical analysis, with a significance threshold was set at *p* < 0.05. A linear mixed model was used to determine the relationship between the LEGSYS+ and MOCAP for the extracted 60 complete gait parameters (20 extracted from each of the 1, 2, and 3 km/h walking speeds). For the model, the point estimate and its 95% confidence intervals were determined using the maximum likelihood function. Adjusted R^2^ of models were determined. As the statistics were performed with MATLAB built-in function “fitlme”, the linear regression was determined using Wilkinson notation [49].
MOCAP ~ LEGSYS+(1|participant)

Agreement between systems was described using Bland–Altman analysis [50]. The standard error of the measurement (SEM) was assessed as the SD of the pooled SD-s within each participant of the difference between the measurement systems (MOCAP—LEGSYS+) divided by the $\sqrt{2}$ [51]. This represents the variation across strides of the difference between the measurement systems. From the SEM, the smallest detectable change can be determined (SDC) [51]; $SDC=1.96\times \sqrt{2}\times \mathrm{SEM}$, and represents the 95% confidence interval; SDC95. This represents the value above which a change in LEGSYS+ is beyond potential measurement error [51]. It should be noted that the above agreement determination assumes that the difference between the measurement systems follows a normal distribution. To test if these assumptions were met, the difference between the two measurement systems was modelled using a linear mixed method fitted to the Bland–Altman plots.
Delta ~ MOCAP+(1|participant)

Were Delta = MOCAP − LEGSYS+

Correlation coefficient interpretations are as follows: negligible (0–0.1), weak (0.1–0.4), moderate (0.4–0.69), strong (0.7–0.9) and very strong (0.9–1) [52].

## 3. Results

### 3.1. Stride Gait Parameters

Each individual extracted gait parameter was explored graphically using data from all treadmill speeds using scatter and Bland–Altman plots (Figure 5 and Figure A1). Limits of agreement and bias (Table 1), of the bland-Altman plots showed a positive bias for stride time and cadence, and a negative bias for stride time and stride velocity. Excellent correlation between the LEGSYS+ and MOCAP for the stride parameters (stride length, stride velocity, cadence, and stride time) is shown (Table 2). In other words, LEGSYS+ was able to predict the MOCAP derived gait parameters very well. The r^2^ values indicate 96–99% of the variation can be explained by the linear relationship, and the slope of the model was very close to 1 (Table 2). However, the upper and lower confidence interval of both stride length and stride velocity were all just below one. This indicates with increase in stride length and stride velocity the LEGSYS slightly (very minor) over-estimates the values (Table 2). A Bland–Altman linear model (Table 3) indicated the error was slightly greater (more positive) at higher values for stride velocity, but lower for stride length.

### 3.2. Phase Gait Parameters

The phase parameters (right and left stride and stance, left and right double support) show a poor relationship between the systems (Table 2), with 33–46% of the variation being explained by the linear relationship (Figure A2, Figure A3 and Figure A4). The upper and lower confidence intervals of all phase parameters were positive, indicating an underestimation. Bland–Altman linear models (Table 3) show a significant positive slope, indicating that there is no consistent overestimation or underestimation between the systems

## 4. Discussion

The following study investigated the agreement between stride and phase gait parameters, obtained using the LEGSYS+ and MOCAP. This work was conducted across three different walking speeds to include participants that are said to reflect frail (2 km/h) and healthy (3–4 km/h) individuals [43,44,45,53,54]. At each speed, 20 strides were taken from MOCAP and compared with the LEGSYS+ at the same time points. 

The relationship between LEGSYS+ and MOCAP was determined using scatter plots with linear regression. The linear model and slopes indicated that stride time and cadence calculated by the LEGSYS+ have a very strong agreement with the MOCAP data (Figure 5). Similar results have been obtained in another study investigating the differences between an accelerometer-based system with MOCAP [55]. While using a linear model to investigate the difference between the systems, a strong relationship was found for stride time (r = 0.97, *p* < 0.01). The results indicate that the LEGSYS+ has near-perfect heel contact detection and therefore can be a useful system to determine a person’s stride time and cadence.

In terms of the LEGSYS+, a positive slope was observed for stride length and velocity indicating a relative overestimation compared to MOCAP. Differences between stride gait parameters within similar systems are also well documented in the literature. For example, the measurement of stride length by an accelerometry system (IDEEA) when compared with a force plate, demonstrated a 7% relative underestimation of stride length (10.8 cm) [56]. Similarly, a study comparing an inertial measurement unit with a developed gyroscope-based algorithm and MOCAP, underestimated stride length by 3 cm, which is much smaller than that of the current study [57]. However, this investigation eliminated strides that were outside of the duration of 1.25 times the median, thus reducing the variability in the strides considered. These are contrasting with the current investigation, with an overestimation of stride length and velocity shown. Similar to the current study, a previous investigation showed when using inertial measurement unit (IMU) methods, that an increase in gait speed saw an increase in anterior-posterior axis velocity of leg sensors, causing an overestimation of stride lengths [58]. Often algorithms for these types of sensors assume the foot and leg have a zero velocity at the time during stance [57]. This is difficult as leg and foot sensors are moving continuously. As the heel strike saw near perfect alignment, differentiating values for the velocity of the leg may be the cause of the stride length and velocity overestimation was seen in the current study. When using LEGSYS+ for absolute measures, this consistent overestimation needs consideration. However, because of the consistent difference, the LEGSYS+ can be considered whilst comparing the absolute difference between populations. In terms of LEGSYS+’s clinical application, the differences observed may not affect its applicability. Despite the differences observed these are not considered clinically problematic as previous studies investigating, for example, major adverse events, indicate a decrease of 0.15 m in stride length as clinically significant [21]. 

Inconsistent differences between the LEGSYS+ and MOCAP were observed for phase parameters with an increase in discordance shown to be dependent on the absolute magnitude of the measurement. It is well documented that gyroscope and accelerometer systems have a much weaker validity and reliability in comparison to other measures in the swing and stance phases [57]. An incongruency between swing duration results (ICC 95% CI: 0.43 (0.07 to 0.69) has also been seen in a study implementing a gyroscopic system (as used in the LEGSYS+) [57]. As explained previously, the algorithm assuming leg velocity as zero causes an overestimation of stride parameters, it concurrently causes an underestimation of the phase parameters such as stance [57]. Accurate toe-off timing is required to determine both stance and double support duration. Methods used to determine motion capture phase parameters use an estimation that is highly correlated with toe-off detection [46] which indicates that the LEGSYS+ toe-off detection is erroneous. Therefore, the findings of this study in conjunction with the preexisting literature suggest that gyroscopic algorithm-based systems require further research if implemented into the community as opposed to laboratory-based MOCAP systems.

The current investigation has the following limitations that require consideration. It has been described that in some instances, extended gait samples of more than 100 strides of testing are required to characterise reliable variability [59,60]. Therefore, future studies investigating higher walking speeds using the LEGSYS+^TM^ may need longer testing periods than 30 s to allow the analysis of more strides. The fact that the current study was performed on the treadmill which has the potential to alter a gait pattern also requires consideration. Therefore, further research characterising the difference between systems on flat ground (mimicking the natural walking environment) and treadmill devices, may require further investigation. Furthermore, the study population included a younger cohort of healthy community-dwelling individuals, which may not be applicable to other cohorts such as individuals with preexisting clinical conditions. Future investigations may consider looking at different placings of the sensors, performing the study on multiple days and including more speeds.

In summary, across both systems, excellent congruency was seen amongst stride time and cadence within a healthy adult community population. Consistent systematic differences in stride length and velocity values were also observed. In terms of stride length and velocity, the findings of this investigation demonstrate that the difference in results must be considered when using parameters for absolute values, but not whilst comparing cohorts using analogous methods. Furthermore, inconsistent differences in phase parameters were detected amongst the LEGSYS+ and MOCAP systems, highlighting a need for further investigation. Despite these limitations, the results of this study indicate that within a community setting for conditions involving gait alterations (i.e., frailty) the LEGSYS+^TM^ is a suitable portable assessment system. However, the results obtained are not directly comparable to those of MOCAP found in a dedicated gait laboratory.

## Figures and Tables

**Figure 1 sensors-23-00537-f001:**
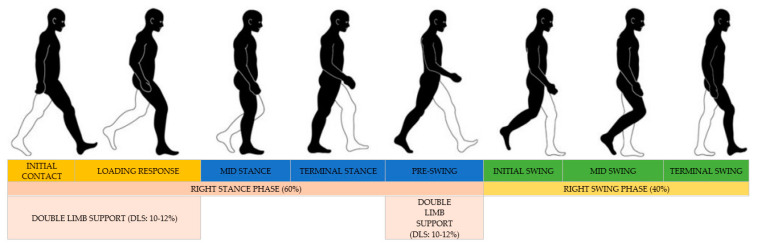
Example of a typical gait cycle (emphasis right leg) with phases, percentages, and functional periods.

**Figure 2 sensors-23-00537-f002:**
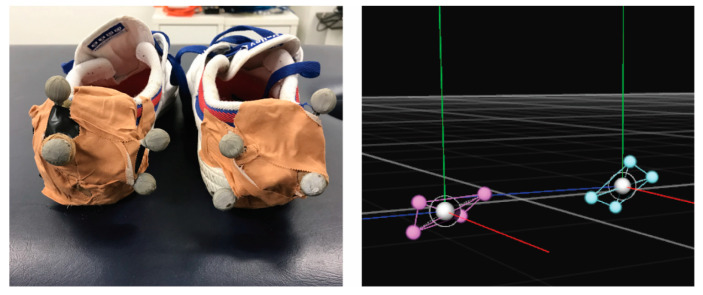
Reflective passive clusters on shoes and representation in the Motive program. (**Left**): A reflective marker cluster was taped to the back of each plimsoll shoe for testing. (**Right**): Marker representation in Motive, with the blue, red and green lines indicating the X, Y, Z axes of the clusters, respectively.

**Figure 3 sensors-23-00537-f003:**
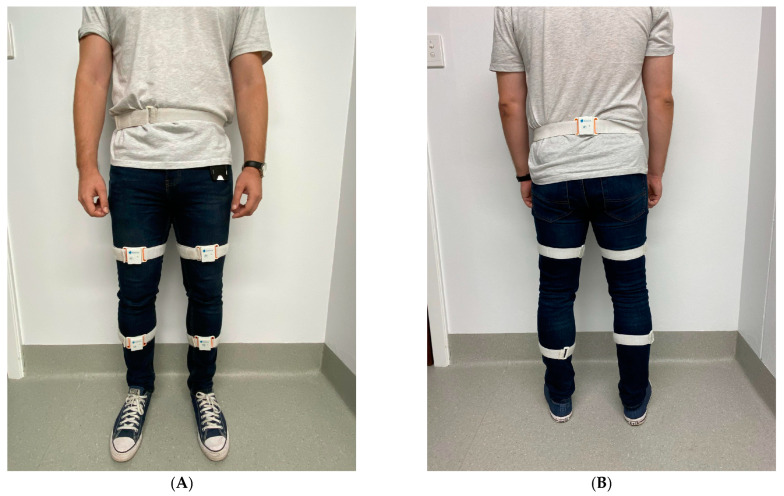
The LEGSYS+ system. (**A**,**B**) show the LEGSYS+ system on a participant, with blue (leg), green (thigh) and orange (waist) being the standardised positions of each sensor. (**C**) Placement of the treadmill with the MOCAP system. (**D**) Storage box of the LEGSYS+ system with each of the 5 Bluetooth motion sensors on display.

**Figure 4 sensors-23-00537-f004:**
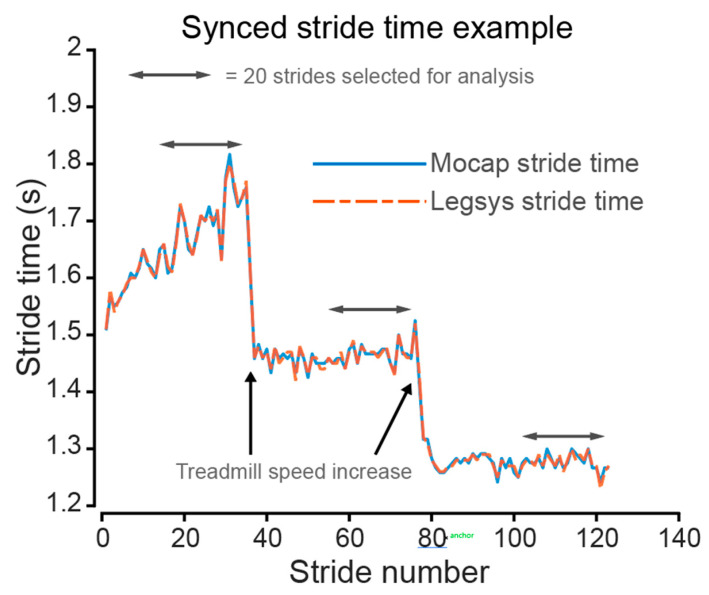
Synchronisation example between the LEGSYS+ and MOCAP for stride time. Plot of stride time against stride number for LEGSYS+ (blue line) and MOCAP (orange dotted line) for a single individual. The single head arrows show treadmill speed changes (2, 3, 4 km/h). The twenty strides prior to treadmill speed change (doubled head arrow) were selected for comparison.

**Figure 5 sensors-23-00537-f005:**
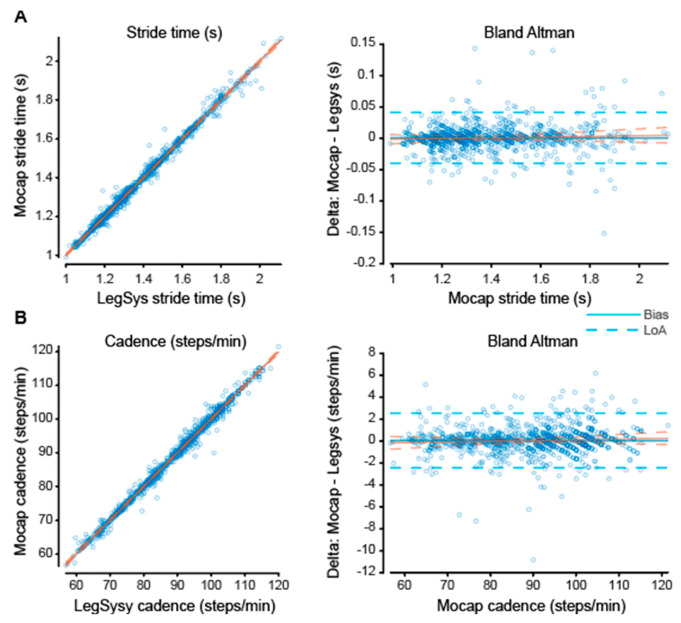
Scatter and bland-Altman plot of MOCAP vs. LEGSYS for stride time (sec) and cadence (steps/min). Part (**A**,**B**) shows the scatter plot and bland-Altman plot on the left and right, respectfully for each parameter. For the scatter plot, the blue dots show stride length or stride velocity for each system. The bland-Altman plots show the difference between the MOCAP and LEGSYS parameters on the Y axis and MOCAP stride length or velocity. Blue line and dotted blue line show the bias and limits of agreement respectfully. The orange line on all 4 plots shows the least squares regression lines with orange dots being the 95% confidence interval.

**Table 1 sensors-23-00537-t001:** Bland altman limits of agreement and bias.

	Limits of Agreement (Low)	Limits of Agreement (High)	Bias
**Stride Measures**			
Stride Time (s)	−0.04	0.04	0.001
Stride Length (m)	−0.22	−0.02	−0.12
Stride Velocity (m/s)	−0.16	−0.01	−0.09
Cadence (steps/min)	−2.43	2.55	0.06
**Phase Measures**			
Left Swing Phase	−7.39	7.49	0.05
Right Swing Phase	−7.14	5.19	−0.97
Left Stance Phase	−7.49	7.39	−0.05
Right Stance Phase	−5.19	7.14	0.97
Left Double Stance Phase	−8.43	7.18	−0.63
Right Double Stance Phase	−5.29	8.43	1.57

**Table 2 sensors-23-00537-t002:** Mocap gait parameters prediction models.

	Intercept (Upper CI, Lower CI)	Intercept *p*-Value	LEGSYS+ Slope (Upper CI, Lower CI)	LEGSYS+ Slope *p*-Value	R^2^
**Stride Measures**					
Stride Time (s)	0.005 (−0.003, 0.01)	0.24	0.99 (0.99, 1)	*p* < 0.001	0.99
Stride Length (m)	−0.08 (−0.1, −0.05)	4.70 × 10^−11^	0.96 (0.95, 0.97)	*p* < 0.001	0.96
Stride Velocity (m/s)	−0.04 (−0.05, −0.02)	3.65 × 10^−9^	0.94 (0.93, 0.95)	*p* < 0.001	0.98
Cadence (steps/min)	0.31 (−0.25, 0.88)	0.28	0.99 (0.99, 1)	*p* < 0.001	0.99
**Phase Measures**					
Left Swing Phase	23.77 (22.07, 25.44)	3.33 × 10^−123^	0.37 (0.33, 0.41)	1.20 × 10^−58^	0.40
Right Swing Phase	20.49 (18.85, 22.14)	3.62 × 10^−102^	0.44 (0.40, 0.48)	1.40 × 10−^80^	0.46
Left Stance Phase	39.52 (36.84, 42.20)	7.30 × 10^−132^	0.37 (0.33, 0.41)	1.20 × 10^−58^	0.40
Right Stance Phase	35.74 (33.16, 38.31)	4.37 × 10^−120^	0.44 (0.40, 0.48)	1.40 × 10^−80^	0.46
Left Double Stance Left Phase	8.25 (7.43, 9.07)	5.43 × 10^−73^	0.30 (0.26, 0.34)	1.75 × 10^−47^	0.39
Right Double Stance Phase	9.24 (8.4, 10.07)	1.27 × 10^−84^	0.34 (0.29, 0.39)	1.33 × 10^−37^	0.33

**Table 3 sensors-23-00537-t003:** Bland altman linear regression data.

	Intercept (Upper CI, Lower CI)	Intercept *p*-Value	LEGSYS+ Slope (Upper CI, Lower CI)	LEGSYS+ Slope *p*-Value	R2	Standard Error of Mean (SEM)	SDC
**Stride Measures**							
Stride Time (s)	−0.007 (−0.016, 0.001)	0.095	0.006 (−0.0003, 0.01)	0.06	0.003	0.02	0.06
Stride Length (m)	−0.14 (−0.16, −0.12)	1.70 × 10^−30^	0.02 (0.003, 0.03)	0.02	0.37	0.04	0.11
Stride Velocity (m/s)	−0.05 (−0.07, −0.04)	3.82 × 10^−16^	−0.04 (−0.05, −0.03)	2.15 × 10^−20^	0.36	0.03	0.09
Cadence (steps/min)	−0.53 (−1.09, 0.03)	0.066	0.007 (0.0003, 0.01)	0.04	0.003	1.27	3.52
**Phase Measures**							
Left Swing Phase	−12.32 (−15.37, −9.27)	6.06 × 10^−15^	0.33 (0.25, 0.41)	4.21 × 10^−17^	0.41	3.00	8.33
Right Swing Phase	−10.70 (−13.41, −7.99)	2.46 × 10^−14^	0.26 (0.19, 0.33)	2.16 × 10^−13^	0.33	2.61	7.25
Left Stance Phase	−20.66 (−25.51, −15.80)	2.30 × 10^−16^	0.33 (0.25, 0.41)	4.21 × 10^−17^	0.41	3.00	8.33
Right Stance Phase	−15.44 (−19.85, −11.03)	1.21 × 10^−11^	0.26 (0.19, 0.33)	2.16 × 10^−13^	0.33	2.61	7.25
Left Double Stance Phase	−4.62 (−6.18, −3.06)	8.09 × 10^−9^	0.33 (0.23, 0.42)	9.23 × 10^−14^	0.38	3.18	8.82
Right Double Stance Phase	−5.32 (−6.63, −4.01)	5.83 × 10^−15^	0.52 (0.45, 0.59)	9.51 × 10^−44^	0.41	2.94	8.14

## Data Availability

Not applicable.

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
