# Peer review of "Comparison of a Wearable Accelerometer/Gyroscopic, Portable Gait Analysis System (LEGSYS+TM) to the Laboratory Standard of Static Motion Capture Camera Analysis"

_sensors, 2023, doi:10.3390/s23010537_

Round 1

Reviewer 1 Report

Introduction:

The introduction is written in great detail and is supported by literature sources.

Perhaps the reason why LEGSYS+TM is interesting for validation could be named in the research question.

Methods:

It would be nice to have pictures of how the sensors are placed or of the test persons on the treadmill in order to better visualise the experimental set-up.

a major problem in such studies is to make the sensor placement the same for all subjects. what measures have been taken to make this comparable?

What sampling frequencies do the individual systems use and how were the data series synchronised?

How and on what basis was the filter frequency of 5 Hz determined?

Results:

The presentation of the results is clear and comprehensible for the reader - there is nothing to add.

Discussion

It would be interesting to have an assessment for which clinical examinations the LEGSYS+TM is suitable as a direct alternative.

Reviewer 2 Report

This work reported a comparison between wearable accelerometer/gyroscopic, portable gait analysis system (LEGSYS+ TM) and the laboratory standard static motion capture camera analysis (MOCAP). The agreement between stride and phase gait parameter was investigated. Consistent systematic differences in stride length and velocity values were observed. In addition, the inconsistent differences in phase parameters were also demonstrated amongst the LEGSYS+TM and MOCAP systems, which needs further investigation. The objective of this work is clear. Considering the comprehensive data and analysis, the manuscript is recommended for publication after minor revision. Some questions and suggestions are as follows:

1.     For the wearable system, the sensor needs to be fixed on different positions of the legs/waist and collect data in real time during body movement, I would like to ask whether the way of fixing the sensors has any influence on the test results? And if you run the same test twice on the same person, will you get the same results and how about of the error?

2.     In Table 2, with increase in stride length and stride velocity the LEGSYS slightly over-estimates the values. What do you think might be the cause? Please try to explain.

3.     In Table 2, the upper and lower confidence intervals of all phase parameters were positive, indicating an overestimation. Please add some explanation about this phenomenon.

4.     The phase parameters (right and left stride and stance, left and right double support) show a poor relationship between the systems (Table 2), with 33-46% of the variation being explained by the linear relationship. Can you give a possible reason for this result? Can a nonlinear relationship be used for comparison here?

5.     What do you think can be done to reduce the error of the wearable systems?

6.     The resolution of all the Figures are not enough.  

Reviewer 3 Report

To enable portable analysis of gait data, the wearable accelerometer / gyroscopic, gait analysis system (LEGSYS+TM) and the standard of static motion capture camera (MOCAP) were used to assess healthy participants walking on a treadmill at three different walking speeds. The results show that LEGSYS+ stride parameters can be used in clinical settings, but the phase parameters need to be used with caution. The experiment was conducted professionally, and the discussion was adequate. Overall, this paper could be published in Sensors journal after all the comments are well addressed.

1. For broad impact, some related works are suggested to be referred to in the introduction part, i.e., ACS Appl Mater Interfaces 2021, 13, 15572-15583; Adv Compos Hybrid Ma 2022, 5, 1939-1950.

2. What are the components of portable gait analysis system in the manuscript? How exactly was it assembled? It is suggested to provide corresponding photos and diagrams.

3. Please explain in detail how the portability of the system is achieved in the manuscript.

4. In Figure 3, why is there a big difference in peak shape in a velocity interval? Please explain the reason.

5. As mentioned in the Result, the higher the step speed, the greater the error. How to solve this problem in future applications?

6. It is suggested that the speed interval test be more detailed to extend the range of gait analysis system.

Round 2

Reviewer 3 Report

Accept in present form